



# Combination of traditional nutrient load analysis and storm hydrograph separation unveil unexpected patterns in event-driven nutrient export dynamics in a rural headwater catchment.

Lukas Ditzel[1], Caroline Spill[1], Matthias Gaßmann[2]

[1]Department of Hydrology and Substance Balance, University of Kassel, Kassel, 34125, Germany

*Correspondence to*: Lukas Ditzel (lukas.ditzel@uni-kassel.de)

**Abstract.** First flush and dilution are major effects on solute export dynamics during precipitation events in headwater catchments but are hard to predict, even if catchment properties are well known. Normalized cumulative load (NCL) functions have been used to visualize and classify event-based discharge–load relationships, distinguishing between

dilution, flushing, and linear export behavior. This study presents an enhanced version of the classical NCL function approach by combining it with hydrograph separation. Over an 18-month period, discharge and solute concentrations were monitored in an agriculturally influenced headwater catchment in the German low mountain ranges, with a focus on nitrate ($NO_3^-$) and total phosphorus, and a complementary dataset of major ions. Discharge was separated by using stable water isotope signals into event water and total discharge. Both discharge components were then analyzed for

solute loads (NO3−, total phosphorus, and major ions). The results reveal significant differences in solute export dynamics between event water and total discharge, including unexpected similarities in the export patterns of nitrate and phosphorus. The proposed method also highlights a shift from predominantly linear export behavior in the total discharge (coefficient of variation = 0.13) to more pronounced first flush or dilution patterns in the event water (coefficient of variation = 0.36). These findings indicate a fundamental difference between the discharge processes

governing the solute export dynamics of the catchment. While the signal of total event discharge indicates linear behavior, the separated event water exhibits strong flushing or dilution tendencies, likely linked to the activation of drainage systems and draining of NO3− legacy storages. The proposed method is straightforward to implement, yields statistically robust results for the dataset and provides new insights into solute input pathways in headwater catchments.

## 1 Introduction

In recent years, the development of new methods for tracking the nutrient cycle in headwater catchments has seen great improvements (Vaughan et al., 2017; Zimmer et al., 2019; Peter et al., 2020). The advent of high-frequency in-situ measurement techniques now allows for the continuous monitoring of nutrients and solutes in stream discharge under both baseflow and stormflow conditions (Werner et al., 2019; Richards et al., 2021; Ritson et al., 2022; Spill et al., 2023,

2024). Difficult questions like the quantification of nutrient sources or the detection of varying input pathways are now



more approachable. Nutrient pollution from nitrate ($NO_3^-$) and total phosphorus ($P_{tot}$) remains a pressing question to both ecosystem health and water quality (Smil, 2000; Seitzinger et al., 2005; Cassidy et al., 2011; Mekonnen et al., 2017; Richards et al., 2021; Weitzmann et al., 2022) Recent research has shown that headwater catchments are significant contributors to the nutrient export of river networks, exhibiting highly variable export dynamics during precipitation

events, often only detectable by high-resolution sampling (Jarvie et al., 2012; Wade et al., 2012; Müller et al., 2018). These detailed time series enable the application of advanced analytical tools such as hysteresis analysis, concentration–discharge relationships and normalized cumulative load (NCL) functions (Hathaway et al., 2012; Mussolf et al., 2015; Vaughan et al., 2017; Winter et al., 2024). Those methods provide detailed insights into catchment nutrient export dynamics and are able to capture and quantify effects like the first-flush phenomenon.

The 'first flush' or 'first foul flush' effects are non-linear phenomena in catchments, highly dependent on land use, season, and event intensity (Kincaid et al., 2020; Winter et al., 2024). With the use of NCL functions for load analysis, introduced by Bertrand-Krajewski et al. (1999) and improved by Obermann et al. (2007), the classification of event-driven solute export dynamics in sewer systems and agriculturally dominated landscapes became more accessible. More recent studies have used NCL functions to compare the strength of the first flush between catchments and solutes

(Hathaway et al., 2012) or expanded the method to include pesticides and pharmaceuticals (Peter et al., 2020). Analysis of the NCL function exponent enables assessment of chemodynamic versus chemostatic export behavior (Musolff et al., 2015, Ebeling et al., 2021).

While highly resolved nutrient data is becoming more widely available, identifying the governing hydrological pathways of nutrient export dynamics remains a key challenge (Burns et al., 2019; Ma et al., 2023). A traditional tool for exploring

those pathways is stormwater hydrograph separation based on stable water isotopes as tracer (Blume et al., 2007; Klaus and McDonnell, 2013). It is often applied to separate pre-event from event water (Fritz et al., 1976; Uhlenbrook et al., 2002; Weiler et al., 2003), but has also been modified to differentiate more than two components (Genereux and Hooper, 1998; Klaus and McDonnell, 2013; Semenov et al., 2015). Stable water isotopes act as ideal conservative tracers since they are part of the $H_2O$ molecule and do not adsorb onto soil particles (Clark and Fritz, 1997; Klaus and McDonnell,

55    2013).

Although cumulative load–discharge relationships are commonly calculated, separating discharge components and applying them to NCL function analysis has not yet been explored. This study addresses that gap by combining hydrograph separation with NCL functions analysis, to map solute export dynamics to the separated components. Specifically, we evaluate the export behavior of $NO_3^-$, $P_{tot}$, and major ions ($Na^+$, $Ca^{2+}$, $SO_4^{2-}$, $K^+$, $Mg^{2+}$, $Cl^-$) in both total

discharge and the event water fraction. This approach enables a detailed comparison of solute loads between hydrological flow components during precipitation events and provides new insights into event-scale export dynamics in headwater catchments.



## 2 Materials and Methods

**2.1 Study site**

The Nesselbach catchment (51° 26' 54'' N, 9° 22' 59'' E) is located in the geological region of the 'Westhessische Senke,' part of the Central Uplands. It directly drains into the Esse River, a tributary of the Weser. The Nesselbach has a mean discharge of 8.1 L s⁻¹ and a stream length of 1.8 km from its source to the monitoring station. The monitored catchment area covers 3.2 km² and consists of 65% rural, 25% mixed forest, and 10% settlement/commercial areas. Some

agricultural fields are equipped with drainage systems that directly discharge into the stream. Elevation in the monitored area ranges from 173 m to 306 m, as shown in Fig. 1. Soils are predominantly classified as Pelosol (90%), with brown earth and rendzina found in the forested regions. The underlying bedrock comprises red sandstone and Muschelkalk. Meteorological data were collected from a WMO-compliant weather station (Thies Clima, DL-15) near the catchment area (51° 28' 2'' N, 9° 22' 41'' E) from August 2020 to November 2022. The area has a temperate climate,

with a measured mean annual air temperature of 8.6°C and an annual precipitation of 635 mm.

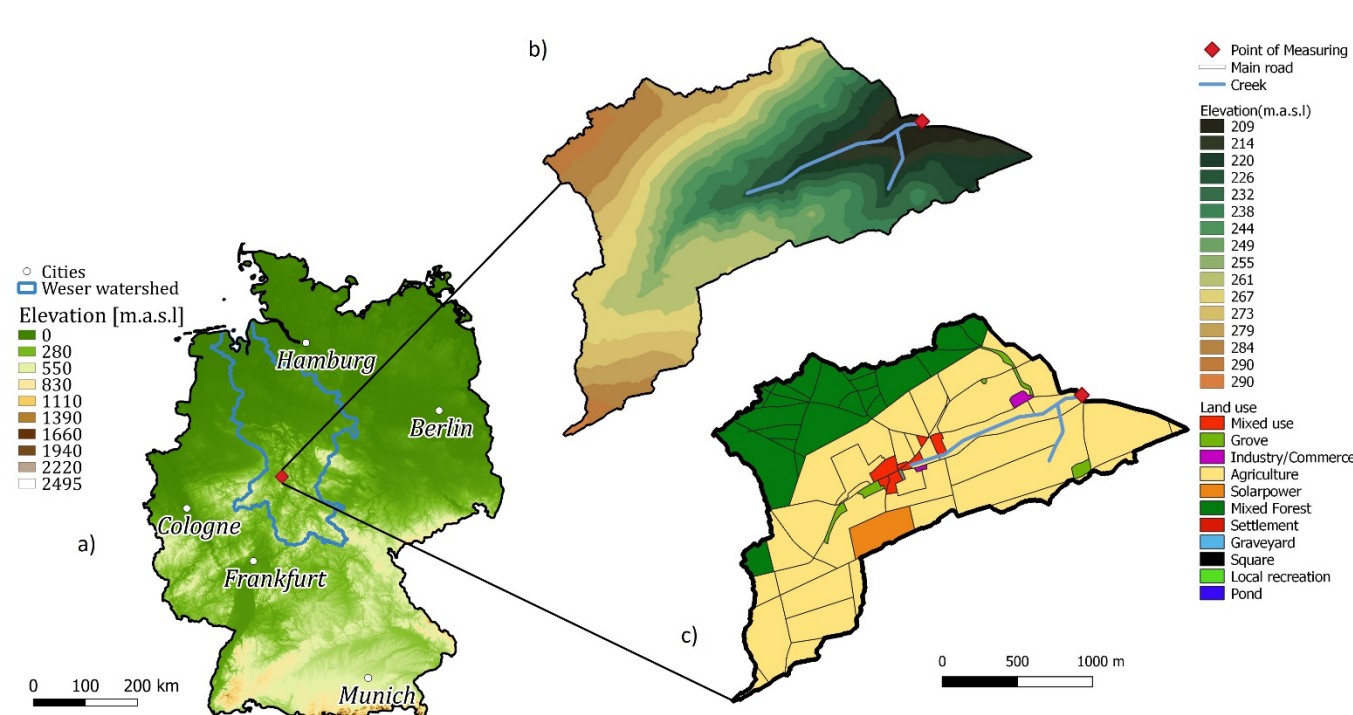

**Figure 1: a) Overview, elevation map of Germany; b) Study area, digital elevation model (elevation in m.a.s.l.), raster cell resolution 1m. c) Study area, land-use, resolution 1:10000.**




## 2.2 Measurement

All necessary parameters, besides climate parameters, were measured at the same location (Fig. 1), where the Nesselbach crosses a road and flows through a 1 m inner diameter concrete pipe. In situ devices were installed at the end of the pipe together with a small weir, ensuring that all devices were always submerged. Nitrate concentration was measured in situ using a UV-VIS probe (S::can Spectro::lyzer) with a temporal resolution of 5 minutes and an automated cleaning procedure, utilizing compressed air. Continuous measurements were complemented by biweekly grab samples for calibration purposes, and additional samples taken during precipitation events. Discharge was determined by measuring water depth and velocity using an ultrasonic probe (Nivus), employing cross-correlation techniques to detect the flow velocity of particles in the water. The temporal resolution of the measurement was set to 5 minutes. Due to the often highly scattered discharge data, caused by low particle density in the water, smoothing of the discharge curve was performed using Spearman correlation between discharge and water depth, followed by applying a Nadaraya-Watson density estimator with a Gaussian kernel to the data. The smoothed discharge was assessed using Nash-Sutcliffe Efficiency, a flowchart is provided in Fig. S 1. Stream water samples during precipitation events were collected using an automatic sampler (ISCO) with a resolution of 15 minutes. The automatic sampler, equipped with 24 bottles (1000 ml), was activated by a water level actuator (ISCO) at an activation threshold of approximately 5 mm precipitation equivalent. Samples were analyzed in the laboratory using UV-VIS photometry for nitrate, major ions and $P_{tot}$ concentrations. Stable water isotope ratios in the discharge were sampled during precipitation events at a resolution of 15 minutes, supplemented by biweekly grab samples. Isotope samples in precipitation were collected using a sequential precipitation gauge with a 5 mm resolution, based on the design by Fischer et al. (2019). A cavity ring-down spectrometer (Picarro L2130-i) was used to calculate isotope ratios. Electrical conductivity was measured every 5 minutes using an in situ probe (HOBO). Major ions were also included into the testing of the new approach. The available data covers major ions for 8 events.

## 2.3 Two component hydrograph separation

The two component hydrograph separation is based on the deuterium signal in precipitation and discharge during events (Clark and Fritz 1997):

$$Q_{pre} = Q_t \left( \frac{\delta_t - \delta_r}{\delta_{pre} - \delta_r} \right) \tag{1}$$

Where $Q_{pre}$ equals the volume of pre-event water, $Q_t$ the discharge at a given timestep $t$, $\delta_t$ the isotope composition in the discharge at a given timestep, $\delta_{pre}$ the isotope composition of the pre-event discharge and $\delta_r$ the isotope composition in precipitation. Due to the sub-hourly resolution of the event samples and the nature of the stable isotopes, it is possible to separate the hydrograph with very high precision. A dual isotope plot (Fig. S 2) was drawn to





identify potential influences of evaporation on the stable water isotopes signal in the precipitation, but the dataset showed no such influence. There were 15 events suitable for stable isotope-based hydrograph separation. These events consisted of at least 6 consecutive discharge isotope samples [15 min time increment] and successfully sampled clearly assignable precipitation samples. For a better understanding of the differences in the event dynamics, the events were

clustered by using k-means cluster analysis, a common and classical method (Lloyd 1982). Cluster analysis can be used to categorize the discharge functions into groups such as fast reacting curves, bi- or multi-modal functions and strength of function volatility (Bloomfield et al., 2015; Wunsch et al., 2022). K-means algorithm is based on the following assumption (Lloyd 1982):

$$J = \sum_{i=1}^{k} \sum_{x_j \in S_i} \| x_j - \mu_i \| \tag{2}$$

Where J equals the target function, k equals the number of pre-identified clusters, x equals the datapoints, μ equals the mean values of each cluster S. The k-means algorithm aims at minimizing the squared distance of the datapoints to the cluster means. After necessary normalization of the data, a Scree-plot (Fig. S. 3) was drawn to visually identify the best number of clusters (Cattell 1966). The k-means algorithm was computed with the R base package (R core Team 2025). Cluster pairs comprised the following parameters: precipitation [mm] and event water portion [%], function mean

volatility [-] and mean rate of change, function mean volatility [-] and area of the best power function fit of the NCL function [-]. The average volatility and mean rate of change of the hydrographs are mathematical parameters delineated from the empirical discharge hydrograph. We calculated these parameters with the sole purpose of grouping the hydrographs by their mathematical behavior. Comparing the mean rate of change, or the abruptness of rising and falling function behavior, is the first step to sorting the hydrograph behavior. To delineate the mean rate of change of a

discharge function (Fig. 2), the first derivative of each discharge function $k$ was calculated and its values averaged to compare the rate of change between the discharge functions $f(Q)$:

$$k = \frac{d}{dt} \, f(Q) \tag{3}$$

With k equals the value of the first derivative of the discharge function at a given timestep and $\frac{d}{dt}$ equals the first derivative of the discharge function $f(Q)$ at timestep $t$. Combined with the calculation of the average first absolute

derivative, which describes the average volatility of a function, it is possible to get a detailed insight into the dynamics of the catchment's hydrology. The average volatility of a hydrograph can be interpreted as the average length of the path the function is taking from the start of an event to the end (Fig. 2):

$$k_{abs} = |\frac{d}{dt}| \, f(Q) \tag{4}$$



With $k_{abs}$ = absolute derivative of $f(Q)$.

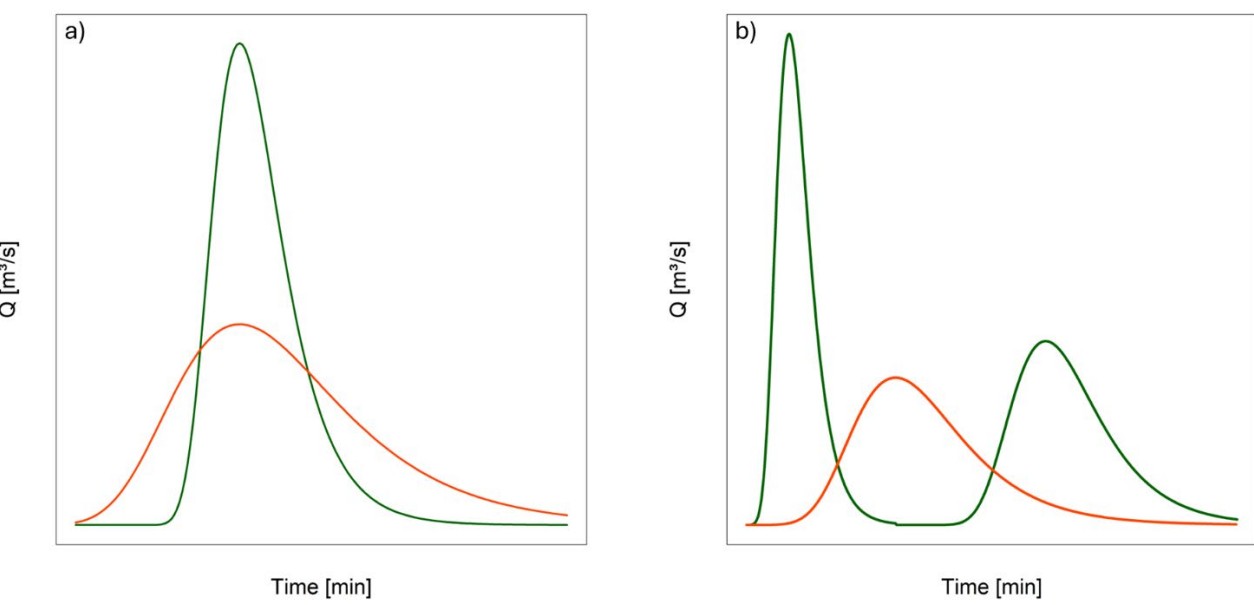

**Figure 2: a) Example for mean rate of change, the green function shows high mean rate of change, the red function shows lower values. b) Example for function volatility, the green function shows high volatility compared to the red function.**

Clusters of separated discharge functions are finally compared by a number of corresponding variables likely influencing event discharge dynamics: total precipitation [mm], antecedent precipitation index of the preceding 14 days before an event (API$_{14}$ [mm]) and precipitation intensity [mm/h]. Following Köhler and Linsley 1951, API$_{14}$ is inversely calculated by the weighted sum of daily precipitation 14 days before the referred event:

$$\text{API}_{14} = \sum_{i=1}^{m} r^i * P_i \qquad (5)$$

With recession coefficient r [-] being 0.9 for Germany as proposed by Schröter et al. (2015), P = precipitation [mm], m = time span (14 days). API is considered as proxy for the pre-event wetness conditions in a catchment and therefore impacts runoff generation during events (Blanchard et al., 1981).

**2.4 Cumulative load analysis**

Event water and total discharge were analyzed for their relative contributions to nitrate, P$_{tot}$ and Major Ions (Ca, Mg, K, Cl, Na, SO$_4$) loads. Beginning with automatic sampling, event water and total discharge were compared against a bisector line, which represents a perfect linear relationship between cumulative discharge and load (Obermann et al., 2009). Functions above this line indicate a strong influence of rising concentrations on the loads (flushing), while





curves below the bisector indicate lower concentrations relative to the discharge (dilution) (Obermann et al., 2009). This method can be used to determine which factor of the load equation dominates the system during an event. The complete formulation of the function can be found in Bertrand-Krajewski et al. (1998). The functions of the events were fitted to a power law function for better visualization and interpretation (Bertrand-Krajewski et al., 1998). The fitting

procedure is derived from Bertrand-Krajewski et al. (1998) and follows a simple power function fit:

$$F(x) = X^b \tag{6}$$

Where $X$ equals the series of normalized discharge samples and $b$ an exponential factor. The optimal value for b can be found by minimizing the root mean square error (RMSE):

$$RMSE = \sqrt{\sum_{i=1}^{n} \frac{(y_i - \hat{y}_i)^2}{n}} \tag{7}$$

Where $y$ equals the observed and $\hat{y}$ (in our case $\hat{y} = F(x_i)$) the predicted value. RMSE can express values between 0 and infinity, with 0 indicating a perfect fit. Figure 3 visualizes the algorithm as conducted for our analysis.





**Figure 3: Visualization of proposed algorithm. Empirical discharge a) and load b) data are normalized, fitted with a power function and plotted against each other, resulting in c), a normalized cumulative distribution function (NCDF) with a bisector (red line). The color coding, green for a hypothetical event water fraction and black for total discharge, shows different exemplary event behavior.**

The values of the exponential factor $b$ shows the behavior of power functions fitted to total load functions (Bertrand-Krajewski et al., 1997). A $b$ value of 1 indicates a steady function aligned with the bisector (constant concentration), while values greater than 1 suggest functions below the bisector (dilution) and values less than 1 indicate functions above it (flushing). Approaching 0 or infinity reflects increasingly divergent behaviors. Separating event water from



total discharge yields two distinct NCL functions per component, revealing significant behavioral differences through *b* values. To complement the exponent analysis with a more interpretable metric of flow component disparities, we calculated areas under the curve using the antiderivative of Equation (6):

$$F(x) \rightarrow \int_0^1 X^b dx \rightarrow \left[ \frac{1}{b+1} * X^{b+1} \right] \tag{8}$$

An area of 0.5 indicates linearity. Values >0.5 imply flushing, and <0.5 indicate dilution. Areas with very small deviation from 0.5 can be regarded as expressions of quasi-linear curves. Thus, areas under the curve (< 0.5) represent stronger dilution during events and areas > 0.5 indicate first flush events. This calculation was performed separately for total and event water discharge.

Correlations between nutrients in separated discharge and total discharge were calculated using Spearman correlation, 185 differences between distributions were quantified using Kolmogorov-Smirnov test (KS test) and t-test ($p < 0.05$ each). For deeper insights, we compared the contrasting behaviors of $NO_3^-$ and $P_{tot}$ during events. Typically, $NO_3^-$ concentrations decrease while $P_{tot}$ concentrations increase due to greater mobilization by fast overland flow in rural catchments (Frazar et al., 2019; Kincaid et al., 2020).

## 3 Results and Discussion

### 3.1 Catchment hydrology

During the monitoring period from 2021-02-01 to the end of summer 2022 the Nesselbach catchment experienced a period of relatively low precipitation (635mm/a) compared to the multi-annual mean of 792 mm/a (DWD 2024). Most discharge events were characterized by total precipitation between 6 and 14 mm, and durations between 15 and 23 hours (Tab. 1). Event 10 was an outlier, with more than four times the discharge of the next highest event (event 15).


**Table 1 Overview of recorded events, sorted by running index. Total P [mm] = total precipitation per event, Mean sep Q [%] = average portion of separated event water.**

| Event | Date | Duration [h] | Total P [mm] | Mean sep Q [%] | Sample size |
|---|---|---|---|---|---|
| **1** | 2021-04-11 | 23 | 13,4 | 0.35 | 24 |
| **2** | 2021-05-10 | 23 | 13 | 0.25 | 24 |
| **3** | 2021-05-18 | 13 | 2,9 | 0.39 | 8 |
| **4** | 2021-05-19 | 14 | 6,7 | 0.36 | 13 |
| **5** | 2021-06-09 | 20 | 6,5 | 0.56 | 11 |
| **6** | 2021-06-24 | 15 | 10,7 | 0.48 | 15 |



| 7 | 2021-06-30 | 30 | 12,5 | 0.44 | 21 |
| 8 | 2021-07-05 | 20 | 4,2 | 0.44 | 11 |
| 9 | 2021-07-08 | 20 | 16,6 | 0.77 | 11 |
| 10 | 2021-07-14 | 62 | 67,4 | 0.08 | 20 |
| 11 | 2021-11-06 | 23 | 6,3 | 0.85 | 7 |
| 12 | 2022-01-04 | 4 | 13,6 | 0.23 | 9 |
| 13 | 2022-02-01 | 27 | 16,2 | 0.45 | 21 |
| 14 | 2022-02-20 | 22 | 10,4 | 0.1 | 12 |
| 15 | 2022-05-20 | 19 | 18,9 | 0.35 | 20 |

The mean event precipitation sum was 14.6 mm, and the mean event water portion was 41%, ranging from 8% to 85%.

K-means cluster analysis was performed on the percentage of event water and precipitation sum (Fig. 4 (a)).

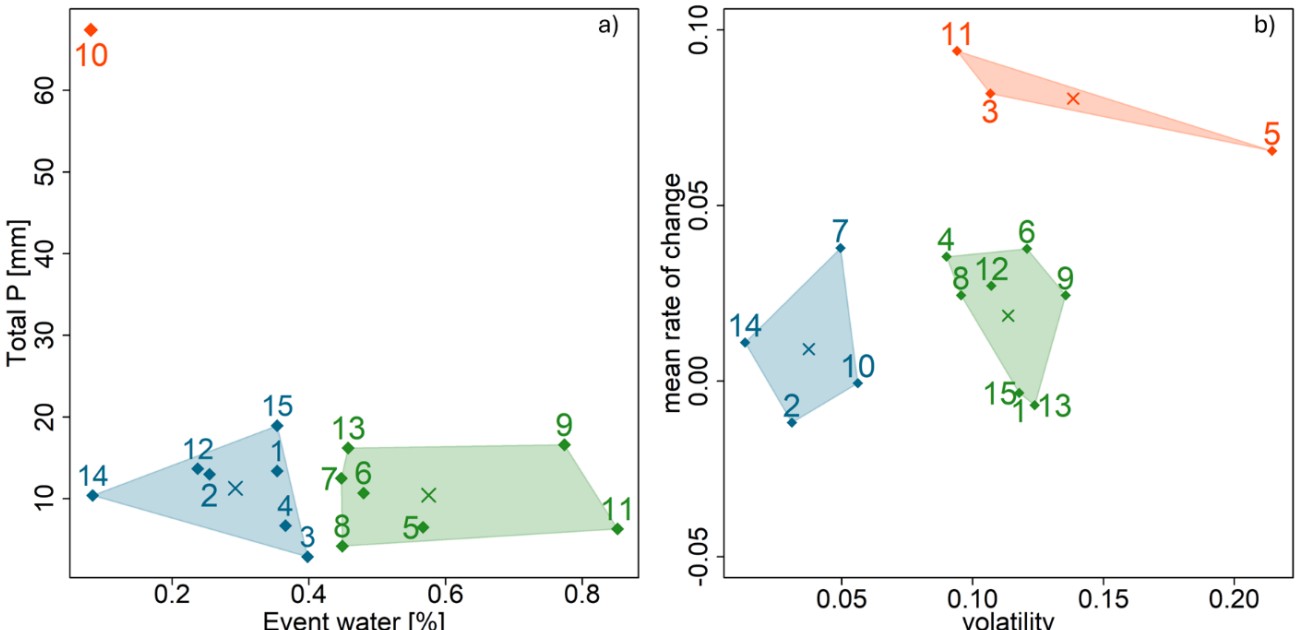

**Figure 4: a) Visualization of K-means cluster analysis of total precipitation per event [mm] vs. mean event water portion [%], clusters are colored. Numbers stand for index numbers of events, see Tab. 1 for details on each event. Centroids of each cluster are marked with x. b) K-means Cluster analysis of mean volatility of each separated event water function and**
**mean rate of change in the relative amount of event water. See Supplement (Fig. S 4) for a visualization of discharge curves per cluster.**

K-means cluster analysis identified 2 clusters for this combination. Event 10 can be considered an outlier with 60 mm

precipitation. The first cluster (blue) comprises 7 events and has its centroid at 10 mm precipitation and 30% event

water. The cluster ranges from very low event water portions (<10%) to moderate portions (>35%), not exceeding the




value of 50% mean event water during any event. The second cluster comprises 7 events with the centroid at 9.5 mm precipitation and 58% event water. Event water ranges from 45% up to 85%. While the majority of datapoints are found in the region between >35% and <50% event water (8 datapoints), both clusters contain data further away from their centroids. The same range of precipitation can lead to discharge events with mean event water portions either

well above or well below 50%. While the blue cluster covers events with low to moderate precipitation and low to moderate event water fractions, the green cluster covers events with nearly the same range of precipitation and moderate to very high fractions of event water. For example, event 11 had the highest event water fraction (85%), despite low precipitation (6.3mm). Pre-event wetness in the catchment could serve as an explanation for this phenomenon, by activating additional flow paths for the event water or enabling significant surface runoff (Duncan et

al., 2017). Figure 5 supports this assumption by showing that greater fractions of event water (green cluster) are accompanied by higher median values and a wider range of $API_{14}$ and a smaller range in max precipitation intensity and event duration.

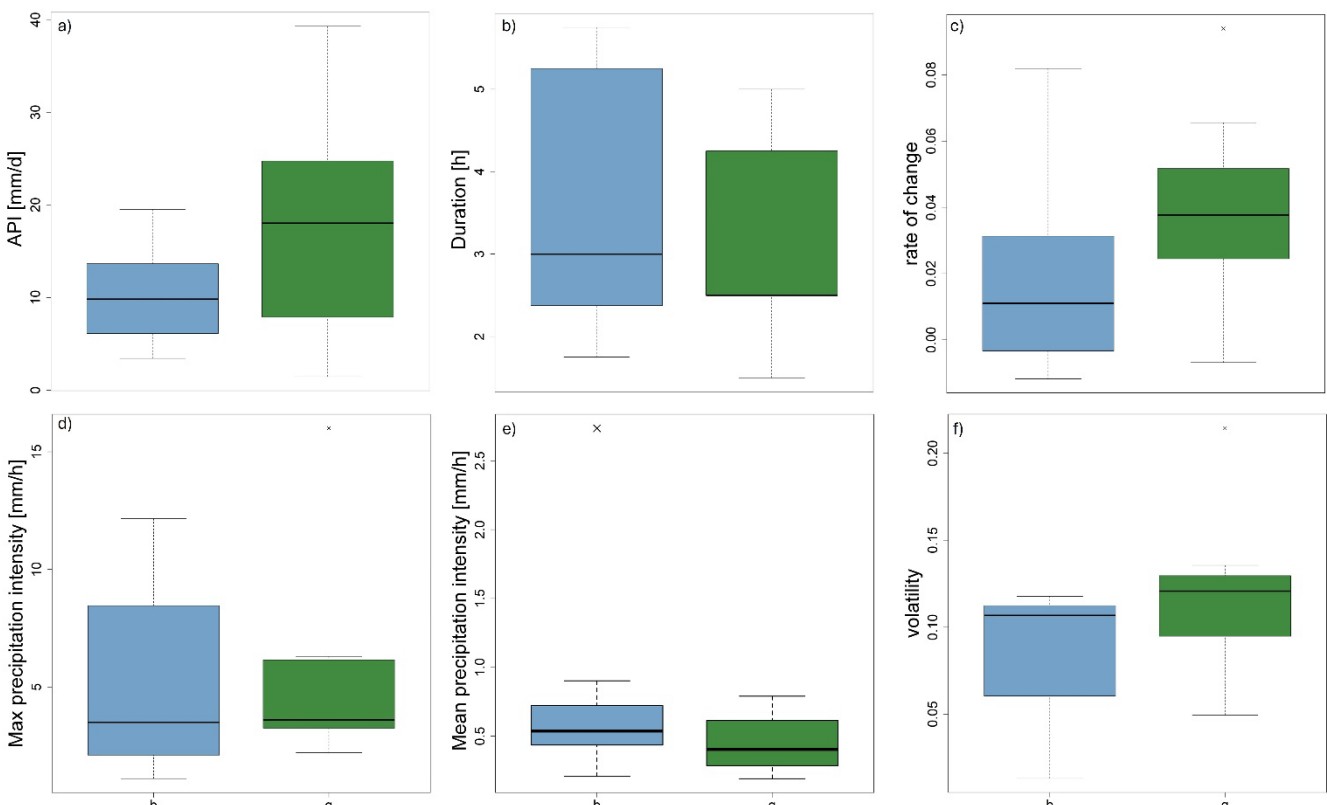

**Figure 5: Boxplots of hydrological parameters for each cluster of Fig. 4 (a), x-values correspond to the cluster color (blue,**
**green); a) API [mm/d]; b) event duration [h ]; c) mean rate of change [-]; d) maximum precipitation intensity in one hour**
**[mm/h]; e) mean precipitation intensity [mm/h]; f) volatility [-].**



In contrast, smaller portions of event water show a smaller range in $API_{14}$ but longer event duration and greater maximum precipitation intensity. Events with smaller event water portions also show smoother discharge curves, as indicated by lower volatility and rate of change. Longer, high-intensity events may mobilize more pre-event water, diluting the event water fraction. Higher precipitation may lead to higher catchment connectivity and may lead to a stronger groundwater celerity wave and therefore a smaller fraction of event water which is mixed with the older groundwater. During shorter events with lower intensity most of the precipitation is quickly entering the stream, leading to high mean event water portions for an event. These observations suggest that the Nesselbach catchment is a system with fast reacting discharge dynamics sensitive to pre-event wetness ($API_{14}$). To sufficiently explain the observed differences in the behavior of separated event water functions, a second analysis was conducted by clustering the mean rate of change (Eq. 3) and volatility of the separated discharge functions (Eq.4) (Fig. 4 (b)). Three clusters were identified by the k-means clustering. The blue cluster contains events with low volatility and moderate rate of change, indicating overall low function dynamics. Two additional clusters were identified, the second cluster (green), which consists of 8 events, shows moderate function volatility at moderate rate of change in discharge. The third cluster (red) consists of only 3 events and shows a high rate of change and moderate to high volatility. On an interpretational level, the red cluster contains events that exhibit fast rising discharge and multiple amplitudes or waves. In contrast, the blue cluster consists of events that show only one amplitude and a slow but steady rise or recession in discharge. Most events fall into the green cluster that shows a steady rise or recession in discharge with more than one amplitude. The statistical interpretation of the dataset is limited due to the small sample size of 15 events and the different length of each event. Groups of functions, corresponding to the results of the k-means cluster analysis, are depicted in Fig. S 4. This behavior of differing function dynamics may be attributed to multiple factors like the precipitation sum, land cover, climatic parameters like temperature and pre-event wetness and the soil type and geology of the catchment (Outram et al., 2016; Peter et al., 2020). It is not uncommon for small catchments to exhibit fast reacting discharge behavior (Alexander et al., 2007; Birkel et al., 2011), but it is necessary to classify the events individually for any monitored catchment to get deeper insights into the nutrient export dynamics. In general, event discharge in the Nesselbach catchment is not mainly governed by precipitation sum but by a variety of parameters.





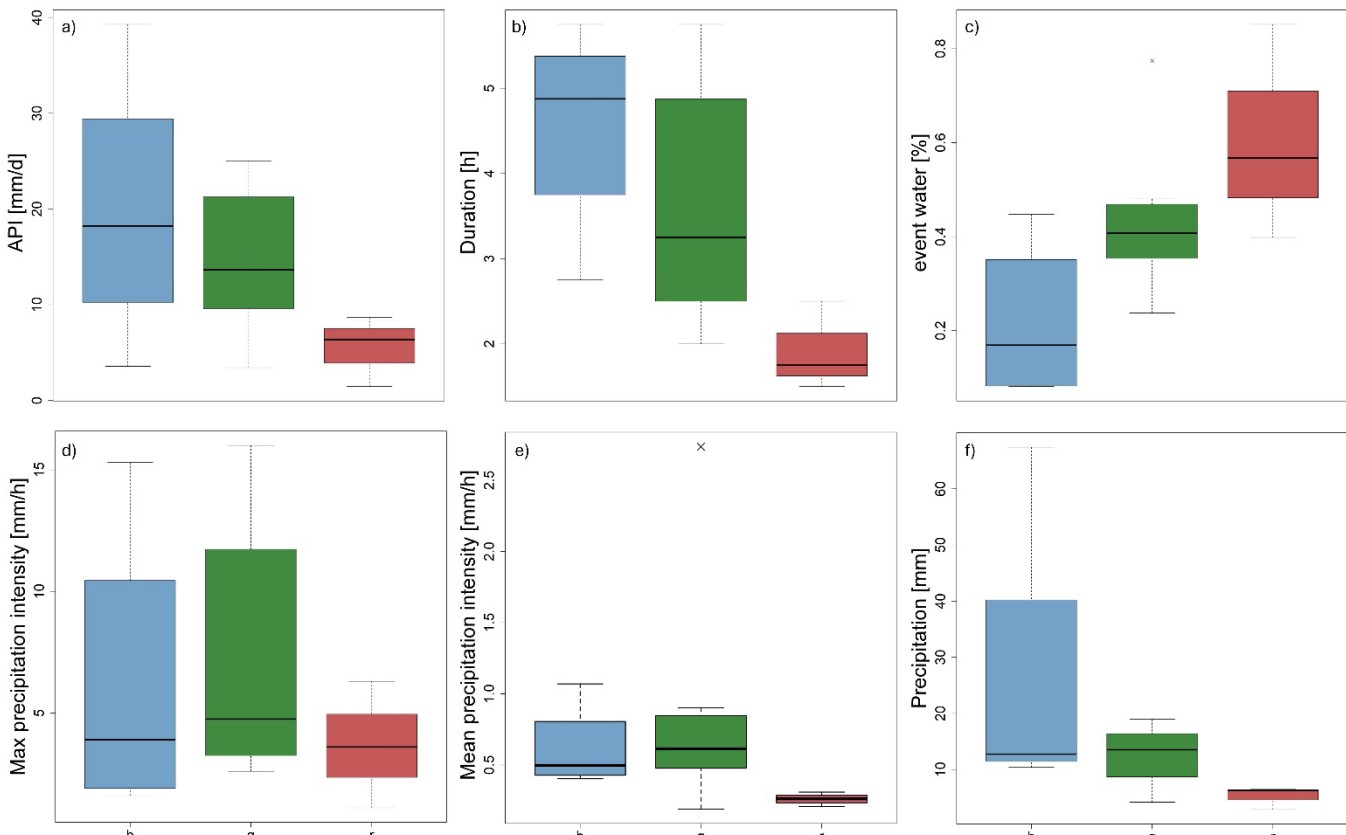

**Figure 6: Boxplots of hydrological parameters for each cluster of Fig. 4 (b), x-values correspond to the cluster color (blue, green, red); a) API [mm/d]; b) event length [h ]; c) mean event water [%]; d) maximum precipitation intensity in one hour**
255 **[mm/h]; e) mean precipitation intensity [mm/h]; f) total amount of precipitation per event [mm].**

 For example, the blue cluster (lowest volatility and mean rate of change) exhibits the widest range for the parameters

API, mean precipitation intensity and total precipitation (Fig. 6). It shows a much wider range for precipitation amount,

than the other two clusters. Events with low volatility and mean rate of change, which translates to smooth and steady

discharge functions, are therefore not necessarily bound to specific precipitation rates or pre-event wetness conditions.

260 This is in stark contrast to the red cluster, which shows the highest mean rate of change and volatility of discharge

functions and expresses much smaller parameter ranges compared to the other clusters. Such dynamic events with

high event water fractions in the monitored catchment require the alignment of several factors. The precipitation and

event duration need to be low, pre-event wetness, approximated by $API_{14}$, shows very low values, and event water

fraction is relatively high. During short rainfall events, precipitation may not last long enough to saturate the soil,

265 resulting in rapid delivery of water to the stream. This effect is likely amplified by seasonal compaction of agricultural

topsoils, which reduces infiltration capacity and enables infiltration-excess overland flow (Kirkby et al., 2014; Peñuela

et al., 2016; Stewart et al., 2019). Combined with the spatial heterogeneity of the catchment, this results in waves of





event discharge and therefore a high rate of change. This assumption is supported by the expression of ranges of the green cluster. The green cluster shows high volatility but a low rate of change, while showing a small range of total precipitation. The range of precipitation intensity during those events is nearly as high, as for the blue cluster. API values within the green cluster show a moderate range, indicating variable but generally elevated pre-event wetness. The green cluster can be described as events occurring at medium to low API, whilst reacting to wide ranges of precipitation intensity and smaller ranges of total precipitation. Therefore, a medium amount of precipitation is enough to trigger volatile events with moderate rate of change if pre-event wetness is high. The events clustered in the green cluster are volatile because of pre-event wetness and moderate amounts of precipitation, due to pre-event wetness and moderate precipitation. This could be attributed to the activation of additional flow paths and the onset of overland flow later in the events. This second clustering confirms the findings of the first by highlighting that the Nesselbach catchment shows highly dynamic discharge during periods of low API (red cluster), higher API values lead to less volatile function dynamics and smooth, unimodal discharge functions.

## 3.2 Load Analysis

The NCL plots of phosphorus and $NO_3^-$ in the total discharge differ visually from each other (Fig. 7(a) and 7(c)). $NO_3^-$ shows a narrower distribution of fitted curves that are more aligned with the bisector, indicating linear behavior (Winter et al., 2022), because $NO_3^-$ is less likely to be mobilized at the beginning of an event and shows in most cases a very linear export behavior. $P_{tot}$ loads are expressing a wider range of curve distribution, indicating a higher rate of dynamic behavior, with a small overweight on first flush effects. A difference in the pattern of nutrient mobilization was expected and follows the literature (Frazar et al., 2019; Ebeling et al., 2021; Spill et al., 2024). The range of values for the area under the curve of the total discharge power functions is 0.33 to 0.57 for $NO_3^-$ loads and 0.38 to 0.62 for $P_{tot}$ loads. It can be observed that there are more events with functions below the bisector for Nitrate (6) than for $P_{tot}$ (4) (Fig. 7 (a) and (c)).

Applying the NCL analysis to the separated discharge dataset changes the outcome and reveals a different distribution of NCL functions (Fig. 7 (b); (d)). While the relationship between total discharge and $NO_3^-$ load is mostly linear, with little deviation from the bisector (Fig. 7 (a)), the NCL functions for the separated discharge show higher distances from the bisector (Fig. 7 (b)), both visual and in more than doubling their range of values for the area under the curve (0.12 to 0.72). These increases in the parameter range in the event water fraction of discharge show a tendency for $NO_3^-$ to express dilution or first-flush behavior that is not observable or way less pronounced in the total discharge. This does not mean that the whole event can be classified as a $NO_3^-$ first-flush or dilution event, but that the transport processes differ between portions of discharge. A linear $NO_3^-$ export behavior in the total discharge can appear as a dilution or first-flush behavior if the NCL curve is using the event water fraction of the same event (Fig. 7). While some events show hardly any alteration, other events are way more sensitive to the observed effect (Fig. 7(a) and (b); Tab. S 1).




**Figure 7: Fitted NCL functions for a) Nitrate in total discharge; b) Nitrate in event discharge; c) P$_{tot}$ in total discharge; d) P$_{tot}$ in event water.**

A similar but less pronounced pattern is also observed for P$_{tot}$. Visually detectable differences in the distribution of curves for P$_{tot}$ are backed by a strong increase in area range increasing from 0.38–0.62 to 0.27–0.77. The Spearman correlation coefficient ($\rho$) for the area under the curve values of the event water fraction of NO$_3^-$ and phosphorus is significant ($p < 0.05$) and positive ($\rho = 0.62$), indicating a dependency of both parameters and a mutual reaction to the precipitation signal. Figure 8 summarizes this analysis by showing the ranges of calculated areas under the curve for the fitted NCL functions of the separated fractions for nitrate and P$_{tot}$ (coefficient of variation = 0.36 and 0.25,



respectively) compared to the total discharge (coefficient of variation = 0.13; 0.14). An overview of the changes in area

for every nutrient and event can be found in the supplement (Tab. S 1)

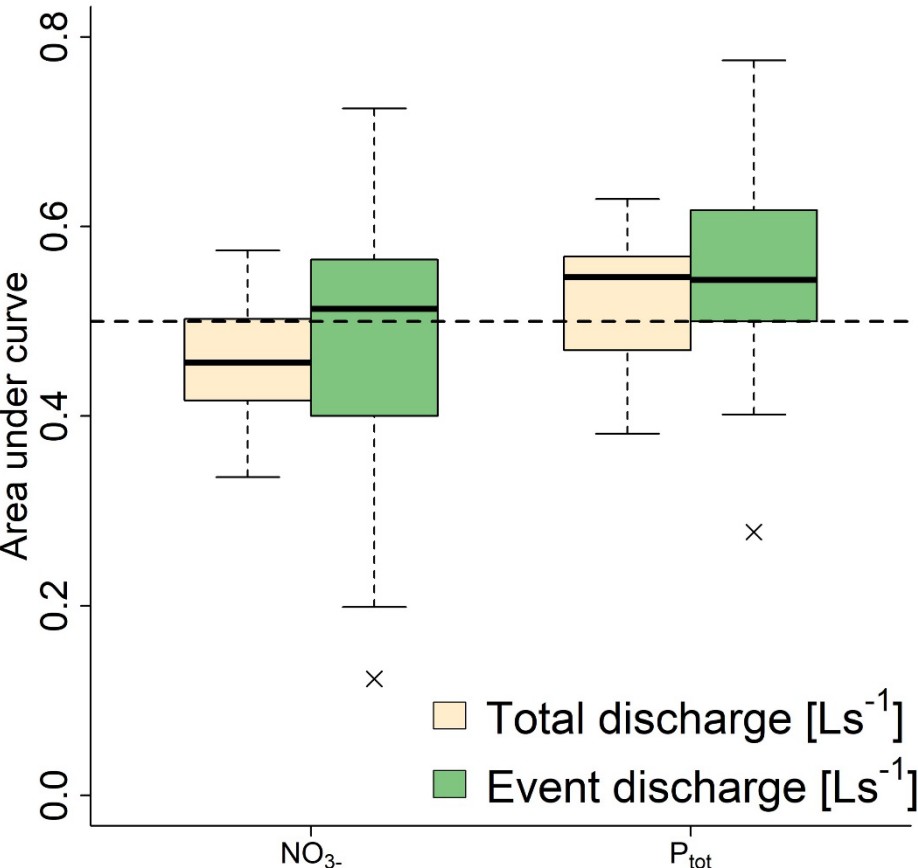

**Figure 8: Boxplots of area under the curve**

The changes in the distribution of NCL functions when using separated discharge instead of the total discharge,

especially for $NO_3^-$ were unexpected but show the potential of the proposed method. Signals of flushing and dilution in the event water were dampened in the NCL functions of total discharge and would have remained undetected without hydrograph separation. To answer the question under which conditions those sudden changes occur, a cluster analysis for the volatility of the separated discharge function of event water and the value of the area under the curve for each nutrient was conducted (Fig. 9).




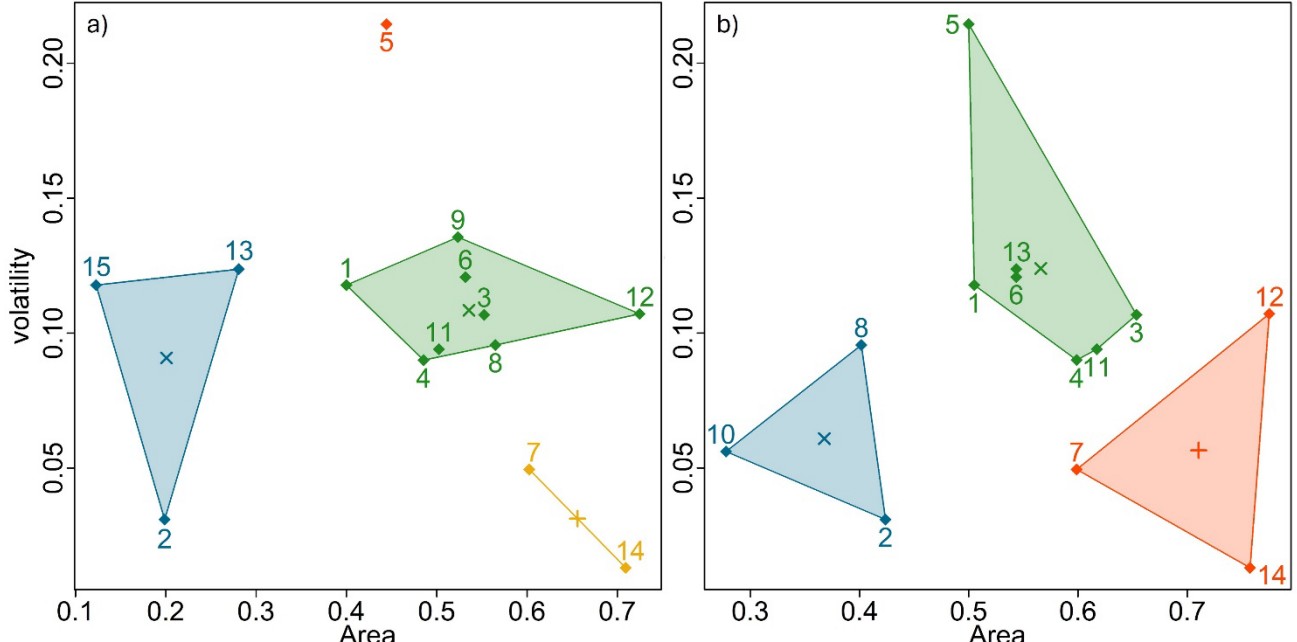

**Figure 9: K-Means cluster analysis for volatility of event water discharge functions and area under the curve of fitted NCL functions per event. a) Nitrate, b) $P_{tot}$. Centroids of each cluster are marked with x.**

Nitrate (Fig. 9(a)) shows three clusters, with the blue cluster clearly showing dilution tendency at moderate volatility, the green cluster showing mostly linear behavior and flushing tendency at moderate volatility, the yellow cluster showing flushing behavior at low volatility and an outlier at high volatility and linear behavior. The pattern of clustering is quite similar for $P_{tot}$ (Fig. 9(b)) but with different events and a less pronounced dilution behavior. Phosphorus loads in the event water express flushing tendency more often. Overall, the range of areas under the curve is smaller for $P_{tot}$ but volatility patterns are quite similar to nitrate. The comparison of the cluster analysis shows that the volatility of event discharge is in some way a governing factor for the expressed behavior of nutrient export dynamics. Flushing and dilution tendencies can be found both during high volatile and low volatile events, while linear export behavior is centered at moderate and even highly volatile discharges. An explanation for this observation can be found in the mechanics of first-flush and dilution: both need a strong signal of precipitation intensity to initialize the effects. The first flush is defined by an overweight nutrient concentration in the beginning of the event and dilution patterns occur when nutrient concentrations are lowered during an event due to an oversupply of water. Highly volatile event discharge is rarely capable of mobilizing enough nitrate to initialize the effect of first flush. A steady supply of event water in the beginning of an event, transporting the nitrate in soil water and overland flow, is needed (Obermann et al., 2009). The same is true for dilution effects: only a steady supply of fresh precipitation water, mostly free of nutrients, can dilute the nutrient background signal. NCL functions aligned to the bisector on the other hand, are not dominated by one of the effects. Due to their more evenly distributed event water discharge, they neither mobilize high nutrient









concentrations nor significantly dilute them. The reasons for the occurrence of first-flush or dilution events are therefore not only attributed to the hydrological functioning of this catchment, but also to the capacity of nutrient storage or seasonally varying effects like pre-event wetness, land use and soil conductivity (Schwientek et al., 2013; Duncan et al., 2015; Winter et al., 2024). This is of great interest, because it was not expected to see such dynamic nitrate export patterns in the event water, which often expresses linear behavior (Winter et al., 2022). Event water is

partly composed of surface runoff, which is often associated with lower $NO_3^-$ loads (Outram et al., 2016; Frazar et al., 2019). These observations contradict the expected behavior of stronger differences between nitrate and $P_{tot}$ loads (Kincaid et al., 2020). The method reveals previously concealed dynamics of nutrient export in this specific catchment. Unexpected nutrient export patterns in the event water might be attributed to catchment properties like the catchments drainage system, unknown point sources, or its legacy storage (mean $NO_3^-$ concentration in discharge = 68

mg/l) (Blaen et al., 2017; Ehrhardt et al., 2019; Spill et al., 2023).

### 3.3 Method validation using major ions

To evaluate the robustness of the proposed method, it was also applied to monitored major ions (Fig. 10). Major ions are suitable for testing the method independently of seasonal nutrient patterns and hydrological conditions.

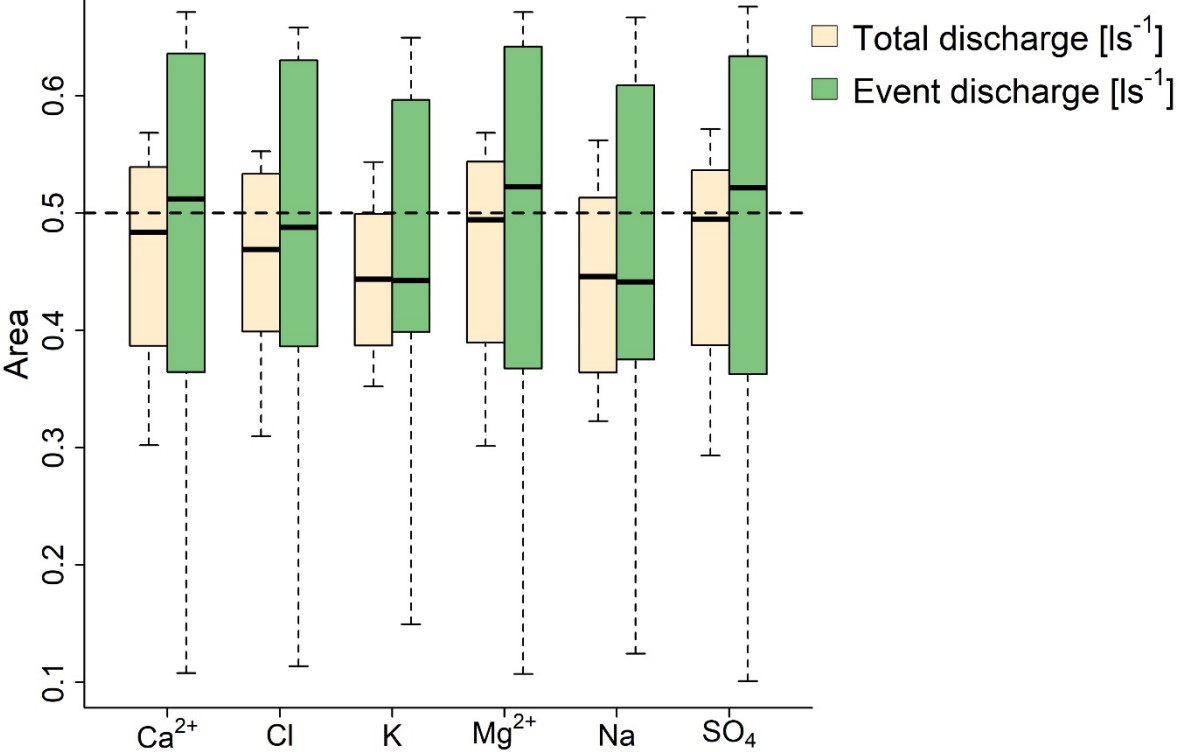

**Figure 10: Boxplots of areas under the curve for each sampled ion species, the dashed line at 0.5 represents linear behavior.**





While the differences between major ions are not as strong as between $NO_3^-$ and $P_{tot}$, differences between event water and total discharge are also prevalent in the distributions of power functions fitted to the ion data. The distributions differ significantly when tested with KS test ($p<0.05$). Potassium expresses the biggest difference between total

discharge and event water and magnesium expresses the smallest. Calcium, magnesium, and sulfate shift from near-linear or slight dilution tendencies in the total discharge to mild first-flush behavior in the event water. The range of calculated areas drastically increases for all ion species (Fig. 10), indicating overall more volatile dynamics of discharge and mobilization of ions in the event water, which is supported by the strong increase in the coefficient of variation for each ion species (Tab. 2).

**Table 2 Coefficient of variation for areas of ions in total discharge and event water.**

| Ion Species | Cv of total discharge | Cv of event water |
|---|---|---|
| $Ca^{2+}$ | 0.2 | 0.4 |
| Cl | 0.21 | 0.39 |
| K | 0.19 | 0.38 |
| $Mg^{2+}$ | 0.15 | 0.34 |
| Na | 0.22 | 0.40 |
| $SO_4$ | 0.18 | 0.38 |

The more pronounced first-flush tendencies in the event water fraction were unexpected, given the generally conservative behavior of these solutes under typical flow conditions (Shanley et al., 2011; Moatar et al., 2017). This observation in the monitored catchment can possibly be linked to an influence of either a frequent overland flow or

temporary activation of the catchments drainage system. Spearman correlation analysis ($p < 0.05$) reveals strong positive relationships between the concentrations of most ion species in discharge. These correlations are likely governed by shared hydrological drivers rather than direct chemical dependencies between ions. As discharge rises, all major ion species are mobilized throughout the duration of an event.

### 3.4 Methodological limitations

Like all data-driven empirical analyses, this method needs a robust sample size from which to draw. 15 events over multiple parameters, as presented in this study, are sufficient to satisfy rigorous statistical thresholds. However, as usual, the more data are available, the more robust the method becomes. At first glance, it appears difficult to interpret the comparison between the total discharge and the event water by using baseline descriptive methods like arithmetic mean, median or standard deviation, because they might show no significant changes. However, statistical methods

like calculating the range of values and coefficient of variation show a clear difference, which indicates a more complex



nature of the observed export dynamics. Using the range of values seems to be the easiest way to highlight differences, but it lacks interpretational power, which could be enhanced by applying a KS test if the sample size is sufficiently large. As an inherited limitation from the NCL function analysis, this method does not allow for interpretation of the influence of event duration and the thresholds for identifying first flush and dilution behavior remain somewhat arbitrary

(Bertrand-Krajewski et al., 1998; Hathaway et al., 2012). Therefore, the method requires a broader framework of supporting parameters and complementary statistical analysis like cluster analysis to gain interpretive power. This method is not a stand-alone tool, and it is not suited to quantify the governing processes of the catchment's nutrient export. Rather, it serves to detect whether changing patterns in the catchment's transport processes occur during events.

**4. Conclusion**

The combination of hydrograph separation with nutrient load functions provides a detailed characterization of catchment export dynamics. Enabled by the high accuracy of hydrograph separation with stable water isotopes and high frequency solute data, the proposed method is capable of detecting differences in solute export dynamics between distinct fractions of discharge. Additionally, due to its general approach, it allows for the comparison of water quality

parameters with widely differing chemical properties and behavior, making it easier to detect unexpected patterns and dependencies. The behavior of $NO_3^-$ in the event water and its resemblance to the $P_{tot}$ export pattern in the study catchment raises new questions about the input pathways of $NO_3^-$ in headwater catchments: Is the $NO_3^-$ export in the event water driven primarily by point sources or by diffuse sources? What is the role of intermittent flow and surface runoff, and are decision-makers able to use this knowledge for effective countermeasures against high $NO_3^-$ loads

during events? Combined with the strong first flush tendencies of major ions in the event water, it can be stated that a closer look at the event water solute export dynamics may unveil solute input patterns previously obscured by the solute export signal of total discharge. The proposed method demonstrates robustness across a spectrum of applicable and relevant water quality parameters. While the approach is straightforward and easy to implement, cluster analysis combined with discharge function classification is one way of evaluating the dataset. Although successful in discerning

nutrient export variations in this specific catchment, its applicability to catchments with varying export dynamics requires further investigation. For broader validation, the method should be evaluated using larger datasets from diverse catchments. Given the method's sensitivity to limited sample sizes, we recommend employing datasets from extended hydrological events or those with high temporal resolution.



**Code and data availability**

Field data and processing algorithms are available upon request from the author.

**Author contributions**

MG conceptualized the research. CS and LD collected the data. LD analysed the data and wrote the manuscript. All coauthors reviewed and edited the manuscript.

**Competing interests**

The authors declare no competing interests.

**Acknowledgements**

The authors acknowledge the support provided by the laboratory of Siedlungswasserwirtschaft at the University of Kassel, as well as the assistance of their research assistants during data collection and laboratory analyses.

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
