# Peer review of "Combination of traditional nutrient load analysis and storm hydrograph separation unveil unexpected patterns in event-driven nutrient export dynamics in a rural headwater catchment."

_EGUsphere, 2025_

## Author Comment (AC1)

Dear anonymous referee #1,

thank you for the very detailed and profound review of our manuscript and the additional recommendations on the literature. We really appreciate your effort and insights and we are looking forward to address and discuss the points you highlighted:

Event water and solute concentrations Export patterns of N and P were analysed for total Q and event-Q. However, it is not explained which concentrations have been assigned to event-Q. Assuming that pre-event Q (or everything that is not event-Q) has a constant concentration, which (if I got it right) was also the underlying assumption for event-Q estimation via the isotopic signal, every change in concentration must be attributed to the event water. This way, even a small increase in the concentration could indicate a much higher concentration in event-Q compared to pre-event Q, especially if event-Q makes only a small percentage of total Q. How was that handled in the analysis? As there is nothing explained in this regard, I assume that the same concentrations as measured in total Q were also assigned to event-Q, and I find this conceptually questionable.

In line with that, I miss a paragraph discussing the uncertainties of the event-Q calculation and what that might mean for the interpretation of results. Also, what happens if water from shallow flow path gets mobilized that has a different isotopic signature? This should at least be discussed.

Further, I strongly recommend reading von Freyberg et al. (2018) on the topic of event water calculation using stable isotopes.

We sincerely appreciate your literature recommendation and your comment on our approach, as it gives us the opportunity to clarify our reasoning in greater detail. We fully agree on the importance of discussing the uncertainties in event-flow (event-Q) calculations using stable isotopes, and we will add a corresponding paragraph to the manuscript. However, we were somewhat puzzled by your general critique of our conceptual framework, so let us clarify our rationale.

In our analysis, we separated the event water from total discharge using stable isotopes and assumed a normally distributed nitrate load in the pre-event water (as was the case during the sampling campaign). Consequently, variations in nitrate load should be attributed to the incoming event water—regardless of whether this portion mainly represents "old" water from the saturated zone or overland flow. Distinguishing between these flow paths cannot be achieved within this setup without direct groundwater or overland flow sampling. Nevertheless, if the nutrient load signal remains stable prior to the event, changes during the event can be attributed to the event-water portion, under the assumption that total discharge (Q) and event discharge (Qe) represent mixtures of pre-event and event water from different sources. This approach is not only widely accepted (Blume et al., 2007; Klaus and McDonnell, 2013; Semenov et al., 2015, Marin-Ramirez et al. 2024) but also the most suitable and logical choice when direct sampling of subsurface or overland flow is not feasible due to resource limitations.

Furthermore, we acknowledge the valuable contribution by von Freyberg et al., who introduced an approach to quantify precipitation-to-discharge fractions to trace streamflow sources. While this is an elegant and insightful technique, von Freyberg et al. (2018) also emphasize that it represents an *alternative* rather than universally superior method. Their study was conducted in a different catchment and at a different temporal resolution. From our perspective, their work delineates the internal correlations between

the  $Q_e/Q$  relationship, antecedent moisture, and other parameters derived from the  $Q_e$  signal. Conceptually, our study pursues a similar objective, but with a focus on the signal of nitrate load, allowing us to examine catchment-scale event dynamics at the same level of abstraction without introducing additional layers of complexity. We believe this conceptual clarity is a strength of our approach, and we see the integration of both frameworks as a promising direction for future research

**1. Explanation of NCL and comparison to other approaches**

While I enjoyed reading the section on how measurements were taken, other parts of the methods lack details that allow the reader to follow how exactly the methods were conducted. This is true for the event-Q calculation, but also for the NCL method. It would further help he readers to clearly distinguish what separates this method from the power law relationship between C and Q (C =  $aQ^b$ ), or the respective L-Q relationship (L =  $aQ^{b+1}$ ), for example. As C-Q and L-Q are, from my experience, more commonly applied, readers need a good reason to be convinced that the alternative method presented here is a good alternative, at least for specific cases or questions.

Thank you very much for pointing this out. From our perspective, the NCL (Normalized Cumulative Load) method is widely applied in hydrology—particularly in the subfield of water quality—for analyzing concentrations and loads of dissolved or suspended substances (see for example Obermann et al. 2007; Hathaway et al. 2012; Mamun et al. 2020). The strength of the NCL approach lies in its visualization of fitted power-law curves within the normalized [0,1] parameter space, allowing researchers to easily detect and interpret differences between water portions, catchments, or events at a low level of abstraction, which makes the method highly intuitive.

In our study, we applied the NCL function to loads, whereas the C–Q method you mentioned typically uses concentrations. We chose loads as a more integrative parameter for analyzing event-scale process dynamics in the Nesselbach catchment. While you correctly referred to the L–Q relationship, this method does not normalize the parameter space. If normalization were applied, the resulting behavior would be equivalent to that observed in our approach.

Moreover, NCL functions are cumulative by definition, whereas the examples you provided are not—this explains the observed differences. As illustrated by your example (L=aQ $^{b+1}$ ) this equation effectively represents the integral of the L–Q relationship when the coefficient a is a positive fraction between 0 and 1. Consequently, our approach focuses on load as an expression of catchment dynamics by implicitly integrating time through the cumulative nature of the NCL function.

**2. b --> 0 (dilution)**

I do not agree with the definition of b à 0 for dilution patterns, as it is currently indicated in Figure 3. I might be mistaken, but why should a negative value of b not be possible? For the C-Q relationship described as  $C=aQ^b$ , a negative b indicates dilution. Translated into loads, it becomes  $L=aQ^{b+1}$ , which means that b<0 implies very strong dilution, so strong that despite discharge going up, loads go down. This can only happen if the baseflow (or pre-event Q) concentration decreases as well, which might be rather unlikely, but it is not entirely impossible. Consequently, it should be  $b \to --> -\infty$  for dilution. I am happy to be proven wrong, but I recommend checking this carefully

Thank you very much for highlighting this point! We believe the misunderstanding arises from differences in the underlying modeling approaches. While you are absolutely correct for formulations such as  $L = aQ^{b+1}$ , negative values of b cannot occur in our case due to the nature of the normalized cumulative load (NCL) functions.

By normalizing the parameter space to [0,1], we mathematically constrain the power function to pass through the fixed points (0,0) and (1,1) This transformation forces the curve to remain within the positive quadrant, effectively shaping the parabolic form into a smooth, bounded curve. Values of b < -1 would correspond to hyperbolic functions that diverge towards infinity and thus leave the normalized parameter space entirely, producing unrealistically large (infinite) areas under the curve. Such behavior would indicate errors in measurements or model setup rather than meaningful parameter estimates.

Even moderate negative exponents (e.g., b=-0.5b) would place the function above the unit square, yielding an integral greater than one ( $\approx$ 2), which again is inconsistent with normalized cumulative data. Since NCL functions are monotonically increasing by definition —each time step accumulates more load than the previous one (or keeps the value if load suddenly drops to zero)—the corresponding fitted functions must also be monotonically rising. In this setup, negative exponents cannot occur.

This does not contradict the general mathematical properties of power functions but rather reflects the specific characteristics and constraints of our normalized model. We appreciate the opportunity to clarify this distinction and will revise the text to make the reasoning more explicit.

**3. Start and end point of events**

I could not find a description of how the start and end points of events were defined. However, I find this important, as this has the potential to severely influence the results, especially with respect to the percentage of event-Q. This needs to be clarified, and its impact on the results should be carefully checked

We will add a description of event delineation in the method section.

**4. Discussion relevance and implication**

I would appreciate to hear a little more about the relevance of the topic. Why does it matter? Yes, these patterns were observed, but what does that imply? This applies to the abstract, the discussion and conclusion.

The method is quite cheap and reliable and the observed patterns can lead to different instructions from decision makers. We will elaborate on that further.

**5. Data availability**

I do not see a reason why data from this study should be available upon request and not uploaded to an open repository. If there is an acceptable reason for that, it should be stated in the

data availability section. Otherwise, I advocate for a transparent and easily accessible provision of the data so others can replicate he presented results

Data can be uploaded to an open repository, this is not against our interest, so far we made good experience with data on request, but there is no reason to not upload the data.

**6. First flush**

I got confused by the use of the term first flush and flushing behaviour in the manuscript. It appeared to me that both were used as a synonym for what in other studies is called an enrichment or accretion pattern, meaning that concentrations increase with increasing discharge. In other cases, it appeared to describe an earlier peak of concentrations as compared to discharge, which one could call first flush, or which others have described by clockwise hysteresis. I might have overread things in this regard, but the manuscript would benefit from a clear definition of what term refers to what and how these different patterns are distinctively characterised via NCL. It would also be good to clearly distinguish "first flush" from enrichment (or flushing?) behaviour, but also distinguishing it from the "first flush" that describes a disproportionally high concentration increase during the first event(s) after a drought (e.g., Winter et al., 2022).

Your absolutely right, thanks for highlighting this! We will apply consistency.

**7. Linear vs. chemostatic**

In the manuscript, constant (or chemostatic) solute dynamics are described as "linear". While I understand that this term makes sense from the perspective of the NCL approach, it is somewhat confusing to readers who are more familiar with C-Q or L-Q relationships in the form of a power law relationship. There, enrichment, chemostasis, as well as dilution are linear in the log-log space. Hence, I suggest using a different terminology.

Thanks, we will revise the wording on this one.

**8. Loads vs. concentrations**

Throughout the manuscript, solute dynamics are often referred to as nitrate or total phosphorus, without indicating whether this is about loads or concentrations. As this makes a huge difference, also in the way results are interpreted, I recommend clarifying this throughout the manuscript.

Thank you for your recommendation, we will clarify this point and make it more consistent.

**Minor comments:**

Title: I suggest either saying "The combination" or "Combining traditional..." with a tendency to the second for brevity reasons. As it is now, it reads a little odd. No point is needed at the end of a title.

Much appreciated, we will consider changing the title.

**Abstract**

I struggle with the causal relationships in the first sentence. Is a first flush or dilution an effect of solute export dynamics? Also, see my major comment regarding the use of the term "first flush".

We will revise the sentence, first flush is of course not an effect of solute export but a phenomenon.

L15: NO3- should be formatted to NO3- throughout the manuscript

We will change it to the correct format.

L19: what are "discharge processes" – I suggest referring to hydrological processes here, if this should refer to transport processes and not biogeochemical ones.

We will rephrase to "hydrological processes"

**Introduction**

L26: "Nutrient cycle"? This sounds like a cycle of biochemical transformations. I assume this should rather be something like nutrient storage and transport within and from catchments?

This is not an uncommon description, but we can adjust it for higher precision.

L28: "nutrient and other solutes concentrations" needs adjustment: it is nutrient/solute concentrations that are measured, and nutrients are solutes as well, not either or.

Maybe the wording is slightly off, but we meant that nutrients are solutes as well.

L32. The point is missing.

Thanks!

L46: Musolff et al. and Ebeling et al. use the exponent of the power law relationship between C and Q. Not NCL, this should be distinguished.

It was not our intention to indicate that Musolff at al use NCL, but that they used the exponent b from the fitting procedure for C-Q analysis, hence a more recent iteration of NCLs. We will clarify that.

L50-51: Include insights from von Freyberg et al. (2018)?

Will be considered

**Methods**

L67: Central Uplands, Germany.

Thanks, will be added.

L72: I suggest referring to the world reference base and not to the German one (i.e., brown earth). I guess it is Cambisol?

We will change it to the international reference base

L75: Please add the year for which the mean was calculated

Will be added (2021)

Fig.1: What are the black lines in the land use map? Are these tile drains or just the borders between polygons? If the latter case, they should be removed. Also, the north arrow is missing

The north arrow will be added. Those black lines are in fact small paths between the acres and in the forest. We can exclude them for a smoother presentation. We also consider to differentiate the agricultural use into meadow and arable land.

L82: besides à except

Thanks!

L87: What is the quality of the calibration? I would like to see a plot comparing sensor measurements and grabs samples with a 1:1 line and R2 or similar in the supplement

This plot will be provided in the supplement.

L93: I appreciate Fig S1. Still I would have liked to see something that gives me an idea of how noisy discharge data was and how it looked like after the correction.

This can be done, we can extend the section in the supplement.

L98: Where was precipitation measured and how? Can this be displayed on the map as well?

Precipitation was measured in the north of the catchment, we can put it in the map.

L104: How does this compare to the methods of von Freyberg et al. (2018)? How were deuterium concentrations in P estimated? As a weighted mean? Also, how was the pre-event concentration estimated? Is it a mean across several values, if so which values?

Deuterium concentrations in P were estimated by using a sequential precipitation sampler based on the design of Fischer et al 2019. Precipitation was collected as weighted mean in 5 mm portions and measured individually. The pre-event concentration was estimated by using biweekly grab samples and samples anticipated before incoming storm events. It is the median value of those samples. We will add this information.

L110-112: Nice!

Thanks!

L115: "using" not "by using" here and elsewhere. It does not really help to know that it is a "classical method", common is enough.

Thanks for highlighting, we will change that.

L123: what is the R base package? If it is an additional package, it should be cited. Otherwise, "computed in R (R core team...)" is enough.

Thanks, changes will be applied.

L125: what is the unit of the mean rate of change?

It's still the discharge volume per time; we will clarify that.

L133: With k equal to

Thanks for mentioning, we change this.

L139 here and elsewhere: "=" should be written out outside of an equation

Can easily be applied

L152: Terminology is not consistent. It should be either "nitrate" or "NO3-". Further, if the minus is added, minus and plus also need to be added to all other ions (e.g. Ca2+, etc.)

Will be changed

L153: "Beginning with" sounds odd. Maybe just: Event water and total discharge from automated sampling were compared...

We will revise the sentence.

Eq.: 6: and F(X) is the concentration? Or the load? Why not say this directly? Also, Q would be the more intuitive abbreviation than X

F(x) is the load, because it is all about cumulative load function. We choose X because we try to explain which changes we applied to the underlying concept of NCL-functions, so we rooted for consistency with Bertrand-Krajewsky (1998).

Eq. 7: and here y is used instead of f(x), right? I recommend using the same (and ideally more intuitive) symbols in all equations.

We will consider to make the symbols more precise, but in this regard, the formula is just following publication standards. Y does not mean f(x) because we don't want to calculate the root mean square error of the function f(x) but from its values. While there is conceptual overlap, that would not be formally correct.

L186-188: This statement is too general. Especially as it is underlain by the citation of two studies that only span a hand full of catchments. If any, a large sample study should be cited here. For example, across Germany, Ebeling et al., (2021) show that N tends towards enrichment patterns (increase in C with increasing Q, due to the mobilisation of diffuse sources with increasing catchment wetness) and dilution P towards dilution patterns (decrease in C with increasing Q), due to the dilution of point sources. This is, if I get it right, the opposite of your statement. Note that Ebeling et al. looked at long-term patterns from low-frequency data, and that patterns between these time scales can diverge (Winter et al., 2024). However, Winter et al. (2024) showed that the tendency towards enrichment or dilution remains the same, only less pronounced, during events.

Thank you very much for providing additional information on the literature. We will revise these sentences and also consider the information in the discussion.

**Results & Discussion**

L192: DWD 2024 à I could not find that reference in the reference section. I am not sure if it would make more sense to name this data source in the method section and remove it here?

Thanks for highlighting, we will add the reference.

Table 1: It would be beneficial to add total Q to the table as well and to specify if "date" refers to the starting date of an event.

This can be done, no problem.

L232: From the literature (von Freyberg et al., 2018) and also intuitively, I would have expected a larger event water fraction during larger storms, as during smaller events, a higher percentage of the water fills up empty storages. The manuscript would benefit from a more detailed discussion on why the results diverge from this and from comparing their results to the literature.

Thank you very much for reading this part profoundly, we just mixed thinks up, during internal review. We will correct the sentences.

L249: e.g. seems to be missing in the citation, as these are just exemplary references

Will be added!

Figure 6 & L257: I guess it is API14?

Correct, will be adjusted

L264-265: see my comment above (L232). Was there a seasonal difference in the events analysed? If not, I am not entirely convinced by this argument.

Sentence will be revised and adjusted (see response on L232).

L283-284: Why is it likely to be mobilised, and why would it "normally" have a chemostatic export behaviour? Winter et al. is a good citation here, as it comes from a study comprising a comparably high sample size. However, the authors showed that event patterns are closer to chemostasis as compared to long-term patterns, not necessarily that all events are chemostatic.

We agree, and we are not assuming that every event is chemostatic, but we will revise and adjust the sentence.

L290-294: I assume this is largely because the event water shows a different dynamic compared to total Q?

Yes, this should be the case since event water is only available at events and total Q is also available during baseflow. Part of the analysis aims at separating event water portion from total discharge, so we assumed that the event water signal has a governing effect on the total Q during events-

L305: I assume Ptot and not phosphorus?

Correct, will be corrected.

L306: Is that necessarily a dependency?

We did not do any causality analysis, but the correlation is quite strong and "dependency" is a correct word to frame the relationship. However, it would take further research to conclude if the parameters are dependent on each other, meaning that  $P_{tot}$  enables  $NO_3$  to behave more dynamically during events, or if they are just reacting in the same way to changing processes. So, we are fine with changing dependency to "robust correlation".

Fig.7: The labels on the right are not needed, as all information is already provided on the left. Maybe lines in the plot could be colored for the different events so that readers can see if the direction of changes remains similar for the same events.

Alright, we can adjust that!

L315: I am not convinced the difference is unexpected (see my comment to L290-294 and my major comment regarding event-Q).

We are convinced, that the difference is unexpected. Please keep in mind, that we did not sample any event water portions independently from the total discharge portion. Without hydrograph separation the transport signal of total discharge would have overwritten the transport signal stemming from the event water.

L344: A deeper look into the literature would show quite a few studies where such patterns have been found (e.g., Dupas et al., 2016; Winter et al., 2021, ...)

Here, we discuss the pattern arising from our method found in Fig. 7. Since this method was not used before, it is also not possible that this pattern has been observed before. We will change the sentence to clarify this.

L332-334: This explanation is needed earlier in the manuscript + additional explanations (see major comments on first flush)

We will change that, thanks for mentioning!

L351: Shouldn't this be introduced in the method section already?

It is introduced in the method section, see line 101 and 102.

Table 2: I recommend adding the ratio  $(CV_c/CV_0)$  here as well. It would enable a nice comparison to studies such as those from (Musolff et al., 2015).

This table does not refer to  $(CV_c/CV_Q)$  ratios, introduced by Musolff et al. Here we just present the coefficient of variation used as a statistical metric for comparing distributions of ion loads in the discharge and linking it to differing patterns for the cumulative load functions.

L370: The drainage system is likely to have a strong impact on the results observed. A deeper discussion on this, also in comparison with other studies with and without such systems, could potentially add much value to the discussion.

Unfortunately, the data on the drainage system in this catchment is very scarce and obscure, we have no reliable information.

L382: What is "sufficiently large"? Can this be specified?

Everything that enables statistical testing across events or sample points without any worries about the statistical robustness of these tests. Usually, the number lies anywhere near 30.

L394-396: And this would not be possible with other methods?

It would of course be possible, but as highlighted, our proposed method is quite easy to use and to interpretate, which makes it a good candidate for comparing study sites.

**References:**

Blume, T., Zehe, E., and Bronstert, A.: Rainfall—runoff response, event-based runoff coefficients and hydrograph separation, *Hydrological Sciences Journal*, 52 (5), 843–862, 10.1623/hysj.52.5.843, 2007.

- Fischer, B. M., Aemisegger, F., Graf, P., Sodemann, H., and Seibert, J.: Assessing the Sampling Quality of a Low-Tech Low-Budget Volume-Based Rainfall Sampler for Stable Isotope Analysis, *Frontiers in Earth Science*, *7*, 244, 10.3389/feart.2019.00244, 2019.
- Hathaway, J. M., Tucker, R. S., Spooner, J. M., & Hunt, W. F.: A traditional analysis of the first flush effect for nutrients in stormwater runoff from two small urban catchments, *Water, Air, & Soil Pollution*, *223*(9), 5903-5915, 10.1007/s11270-012-1327-x, 2012.
- Klaus, J. and McDonnell, J.J.: Hydrograph separation using stable isotopes: Review and evaluation, *Journal of Hydrology* 505, p.47-64, 10.1016/j.jhydrol.2013.09.006, 2013
- Mamun, A. A., Shams, S., & Nuruzzaman, M.: Review on uncertainty of the first-flush phenomenon in diffuse pollution control. *Applied Water Science*, *10* (1), 53, 10.1007/s13201-019-1127-1, 2020.
- Marin-Ramirez, A., Mahoney, D. T., Riddle, B., Bettel, L., and Fox, J. F.: Response time of fast flowing hydrologic pathways controls sediment hysteresis in a low-gradient watershed, as evidenced from tracer results and machine learning models. *Journal of Hydrology*, 645, 132207, 10.1016/j.jhydrol.2024.132207, 2024.
- Obermann, M., Froebrich, J., Perrin, J. L., and Tournoud, M. G.: Impact of significant floods on the annual load in an agricultural catchment in the Mediterranean. *Journal of Hydrology*, 334(1-2), 99-108, 10.1016/j.jhydrol.2006.09.029, 2007.
- Semenov, M. Y. and Zimnik, E. A.: A three-component hydrograph separation based on relationship between organic and inorganic component concentrations: a case study in Eastern Siberia, Russia., *Environmental Earth Sciences*, 73, 611-620, 10.1007/s12665-014-3533-x, 2014.

---

## Author Comment (AC2)

Dear anonymous referee #2,

thank you for your review, we very much appreciate the indications for improving our manuscript.

**Response to detailed comments:**

Line 69: If applicable, please change "rural" to "agricultural area".

We will change the wording to "agricultural area".

Line 81: Please revise the heading, possibly change to "Sampling and measurements" to illustrate the content.

Thank you for highlighting that, we will revise the heading to "Sampling setup".

Lines 124-125: Please check that all parameters are listed correctly and use the same terms as in line 126.

We will check for consistency.

Line 103: Please revise the heading, as this section also covers the clustering of events, not just hydrograph separation. Alternatively, move the second part to a separate section.

Heading will be changed to, "hydrological event analysis".

Line 196/ Table1: The pre-event wetness in the catchment (API) appears to be a very important parameter for the subsequent analysis. Please add this to the table.

Thank you very much for this input, the API will be added to Table 1

Line 215-217: Please revise the sentences to avoid repetition.

Sentence will be revised.

Line 225/ Fig. 5: Add a \* if the clusters are significantly different.

An asterisk will be added.

Line 265: It is mentioned here that the red events might have taken place in seasons when no crops were grown or the fields were compacted. However, based on Fig. 4b and Table 1, the red events (3,5 and 11) took place in May, June and November so I cannot see any clear pattern. Please elaborate on this reasoning in the text, and be careful not to draw conclusions based on a very limited sample size.

Also, the events in Table 1 were not equally distributed throughout the year, which may introduce a bias towards spring and summer samples. Please discuss this briefly.

Thank you very much for your observation, there is indeed a bias towards spring and summer: The measurement took place from 02/2021 till 06/2022 due to institutional restriction regarding our field setup. Unfortunately, the autumn of 2021 was very dry and did not bear any major precipitation event we could sample from. In the winter of 2021 snowfall scrambled the isotope measurement and the small stream was completely frozen, making it impossible to measure discharge. The following winter of 2022 left us with three usable events, so we get a sample bias towards spring and summer.

We will revise our reasoning regarding the influence of crops and season on the event dynamics, especially considering limitations in sample size and resulting bias of the database.

Lines 274-276: Please revise sentence.

Sentence will be revised

Line 349: Could the influence of the drainage system on nutrient export be quantified? Is there any data available indicating the percentage of the catchment area with a drainage system? Similarly, this question could be extended to the crop type in the catchment over the years. In general, more information on the catchment's specific agricultural land use would be helpful in order to understand the possible nutrient export.

Thank you for mentioning these points. Unfortunately, there are no more details about the drainage system available. While we identified 2 clearly visibly drainages, there is no information of how large the drained area is, how often those drainages are activated or which amount of discharge is contributed by the drainages at an event. More detailed knowledge about the agricultural land use in the catchment can only be derived from occasionally field observations, there is no official database containing a timeseries of crop types for every field in the catchment. However, we observed that most of the slopes in the catchment were planted with grain (mostly winter wheat and rye) and the shallower fields near the stream were planted with mostly cauliflower and kale. We aim to present the land use in greater detail for the next iteration of the manuscript, by breaking down the agricultural land into meadows and crop areas, and also possible differentiating among grain and vegetable.

Lines 375-389: Possibly add recommendations for transferring this method to other catchments and sampling campaigns.

We will add some recommendations for inferring the method to other catchments for example the increase of precision coming with a more detailed knowledge about land use and crops types, or practical experience like the cleaning interval of the devices.

Line 399: Do you mean general knowledge or is there a specific point, that you would like to highlight to decisionmakers?

We refer to general knowledge.

Line 407: "given the method's sensitivity to limited sample size": was this tested here?

This was not tested in this approach, but since the core of the NCL function is a statistical fitting technique, it will profit from a bigger sample size.